# Effects of Porcine Whole-Blood Protein Hydrolysate on Exercise Function and Skeletal Muscle Differentiation

**Sun Woo Jin** [1,2,†], **Gi Ho Lee** [1,†], **Ji Yeon Kim** [1,†], **Chae Yeon Kim** [1], **Young Moo Choo** [2], **Whajung Cho** [3], **Jae Ho Choi** [4], **Eun Hee Han** [5], **Yong Pil Hwang** [6] and **Hye Gwang Jeong** [1,*]

1   Department of Toxicology, College of Pharmacy, Chungnam National University, Daejeon 34134, Korea; mpassword@cnu.ac.kr (S.W.J.); ghk1900@cnu.ac.kr (G.H.L.); jykim525@o.cnu.ac.kr (J.Y.K.); chaeyeon05@o.cnu.ac.kr (C.Y.K.)
2   Department of R&D, Jinju Bioindustry Foundation, Jinju 52839, Korea; ychoo@jbio.or.kr
3   R&D Institute, AMINOLAB Co., Ltd., Seoul 06774, Korea; wjcho@aminolab.co.kr
4   Subtropical/Tropical Organism Gene Bank, Jeju National University, Jeju 63243, Korea; chlkoala@naver.com
5   Drug & Disease Target Research Team, Division of Bioconvergence Analysis, Korea Basic Science Institute (KBSI), Cheongju 28119, Korea; heh4285@kbsi.re.kr
6   Fisheries Promotion Division, Mokpo 58613, Korea; protoplast@hanmail.net
*   Correspondence: hgjeong@cnu.ac.kr; Tel.: +82-42-821-5936
†   These authors contributed equally to this work.

**Abstract:** A number of studies have utilized blood waste as a bioresource by enzymatic hydrolysis to obtain amino acids, such as branched-chain amino acids, which can increase muscle mass or prevent muscle loss during weight loss. Although a significantly high content of branched-chain amino acids has been reported in porcine whole-blood protein hydrolysate (PWBPH), the effects of PWBPH on skeletal muscle differentiation and exercise function remain unclear. In this study, we investigated the effects of PWBPH on exercise endurance in ICR mice and muscle differentiation in C2C12 mouse myoblasts and gastrocnemius (Gas) muscle of mice. Supplementation with PWBPH (250 and 500 mg/kg for 5 weeks) increased the time to exhaustion on a treadmill. PWBPH also increased the Gas muscle weight to body weight ratio. In addition, PWBPH treatment increased skeletal muscle differentiation proteins and promoted the Akt/mTOR-dependent signaling pathway in vitro and in vivo. These results suggest that PWBPH can be utilized as a bioresource to enhance exercise function and skeletal muscle differentiation.

**Keywords:** porcine whole blood protein hydrolysates; branched-chain amino acid; skeletal muscle differentiation

## 1. Introduction

When slaughtering livestock, blood is treated as waste. Therefore, the 2.5–3.0 L of blood per pig, amounting to approximately 39,216 tons per year, has a major impact on water pollution [1,2]. Blood, which accounts for approximately 7% of the total weight of pigs, generally contains 75–80% moisture and 15–17% high-quality protein; the protein is composed of albumin, globulin, and hemoglobin [3,4]. Although this blood has the potential to become a valuable bioresource, it is disposed of due to a lack of research on possible industrial uses and a general aversion to blood. A recent study showed that the content of free amino acids, including branched-chain amino acids (BCAAs), such as valine (Val), isoleucine (Ile), and leucine (Leu), can be increased by enzymatic hydrolysis of pig blood [2]. In the human body, the 3 proteinogenic BCAAs Val, Ile, and Leu are included among the 9 essential amino acids (EAAs), accounting for 21% of the total protein content and 35% of the EAAs in muscle [5,6]. Especially, the BCAA Leu plays important roles in the regulation of muscle protein synthesis [7–9].

Increases in strength and endurance are related to muscle metabolism and mass. Pathological conditions, such as sarcopenia, cachexia, sepsis, burns, and trauma can weaken

muscles and inhibit muscle damage repair [10]. The muscle regeneration process is necessary for muscle maintenance and damage repair systems and is precisely regulated by the expression of myogenic regulatory factors (MRFs), such as myoblast determination protein (MyoD), myogenin, and myogenic factor 5 (Myf5) [11,12]. Myf5 is initially expressed after satellite cell activation, and newly formed muscle fibers sequentially express MyoD and myogenin [13]. Previous studies showed that Leu promoted the proliferation and differentiation of murine and porcine myoblasts [14,15]. Leu deficiency was reported to inhibit the differentiation of both C2C12 myoblasts and primary mouse satellite cells via suppression of MyoD and Myf5 expression [16].

The accumulation of muscle-specific proteins is associated with muscle growth, and protein deposition depends on the balance between protein synthesis and degradation [17,18]. Previous studies have shown that nutrition can induce protein synthesis via the PI3K/AKT signaling pathway in rats and rainbow trouts [19,20]. The mammalian target of rapamycin (mTOR) is the main mediator of the PI3K/AKT pathway, which plays an important role in protein synthesis [21]. mTOR regulates ribosomal S6 kinase (p70S6K) phosphorylation as one of the key downstream effectors and ultimately stimulates protein synthesis [22]. Leu promotes protein synthesis via the phosphorylation of translation initiation factors and ribosomal proteins of the mTOR signaling pathway [23,24]. The anabolic action of BCAAs is also involved in the activation of insulin secretion and glucose uptake, which lead to beneficial effects on muscle maintenance and muscle loss [25].

In this study, we examined the effects of porcine whole-blood protein hydrolysate (PWBPH) on exercise function in ICR mice and skeletal muscle differentiation in C2C12 mouse myoblasts and gastrocnemius (Gas) muscle of mice.

## 2. Materials and Methods

### 2.1. Chemicals and Materials

PWBPH was provided by AminoLab Co., Ltd. (Seoul, Korea). Briefly, coagulated blood was collected from a slaughterhouse and crushed using a liquefier (AminoLab Co., Ltd.); it was then treated with 1.2% food-grade serine protease (AminoLab Co., Ltd.) and hydrolyzed at 55 °C for 16 h. After hydrolysis, the blood was incubated with 5% food-grade activated carbon powder with shaking for 3 h. The hydrolysate was filtered three times using filters made of diatomaceous earth powder. The hydrolysis and filtration processes were repeated three to four times until a clear yellow liquid was obtained. The filtrate was sterilized at 85 °C for 30 min and powdered using a spray dryer. Dulbecco's modified Eagle's medium (DMEM), fetal bovine serum (FBS), penicillin–streptomycin, and trypsin were purchased from Welgene (Gyeongsa, Korea). Insulin–transferrin–selenium supplement was purchased from Thermo Fisher Scientific (Waltham, MA, USA). Antibodies against MyoD, myogenin, MYH3, Pax-7, atrogin-1, muscle-specific ring finger protein 1 (MuRF1), p-mTOR, mTOR, p-p70S6K, p70S6K, and $\alpha$-tubulin were obtained from Cell Signaling Technology (Beverly, MA, USA). All other chemicals were of the highest grade commercially available.

### 2.2. Amino Acid Content Determination

To determine the free amino acid content, each sample was diluted 10-fold with distilled water and centrifuged at $3000 \times g$ for 20 min, and the supernatant was filtered using a 0.45 μm membrane filter (PVDF-2545; Chemco Scientific, Osaka, Japan) and examined using an amino acid analyzer (L-8900; Hitachi, Tokyo, Japan). The reagents used in these processes were all HPLC grade. To determine the amino acid contents, 15 mL 6 N HCl was added to approximately 0.25 g sample in an ampule. The air was replaced with N2 gas and quickly sealed. The sample was hydrolyzed in an oven at 110 °C for 24 h, allowed to cool, filtered through a 0.2 μm membrane filter, derivatized using the AccQ-Tag method, and then analyzed using an amino acid analyzer (Table 1).

**Table 1.** Physiological amino acid compositions of PWBPH.

| | List of Amino Acid | Contents (mg/100 g) |
|---|---|---|
| Physiological amino acid | Threonine | 1197.01 |
| | Cysteine | 70.79 |
| | Tyrosine | 1111.59 |
| | Arginine | - |
| | Alanine | 2056.28 |
| | Proline | 165.18 |
| | Lysine | 1464.38 |
| | Histidine | 984.23 |
| | Isoleucine | 611.30 |
| | Leucine | 6016.50 |
| | Methionine | 401.51 |
| | Phenylalanine | 2147.77 |
| | Tryptophan | 76.33 |
| | Valine | 1738.06 |
| | Glutamic Acid | 1850.13 |
| | Aspartic Acid | 1520.94 |
| | Serine | 1226.45 |
| | Glycine | 340.33 |

### 2.3. Animals and Treatment

Eight-week-old male ICR mice were obtained from Daehan Biolink (Seoul, South Korea) and acclimatized to the experimental facility for 5 days. The animals were randomly divided into 4 groups (*n* = 6 per group) as follows: control group (saline), 2 groups of PWBPH at different doses (250 or 500 mg/kg), and Leu group (500 mg/kg). PWBPH and Leu were dissolved in saline and were administered via oral gavage once daily for 40 days, while the control group was treated with an equal volume of saline as a vehicle by oral gavage. Mice were housed in a controlled environment (22–23 °C, 12/12-h light/dark cycle) in accordance with the guidelines of the Chungnam National University Animal Ethics Committee (202012A-CNU-191). At the end of the experiment, all animals were fasted for 12 h, and the Gas muscles were collected and weighed.

### 2.4. Exercise Performance

To examine the effect of PWBPH on the exercise performance of the animals, the forced treadmill test was performed using a Touchscreen Treadmill (5-lane treadmill, Panlab/Harvard Apparatus, MA, USA). In the forced treadmill exercise test, the speed of the treadmill was increased by 5 cm/s to 55 cm/s every 3 min and then maintained at this speed until exhaustion, which was defined as the inability to run for 10 s. Animals were trained on the treadmill at days 38 and 39, and exhaustion time was calculated at day 40.

### 2.5. Cell Culture and Myotube Differentiation

C2C12 mouse myoblasts were obtained from the American Type Culture Collection (Manassas, VA, USA). Cells were cultured in DMEM and maintained at 37 °C in a humidified incubator under 5% $CO_2$. C2C12 cells were cultured in growth medium (GM) containing DMEM supplemented with 10% FBS, 2 mM l-glutamine, 100 U/mL penicillin, and 100 μg/mL streptomycin up to 70% confluence. During proliferation, the cells were seeded at $2 \times 10^5$/well in 6-well culture plates and grown in GM. When the cells reached approximately 90% confluence, the GM was removed and replaced with a differentiation medium (DM) consisting of DMEM supplemented with 0.5% FBS and insulin–transferrin–selenium. DM alone or DM with PWBPH (50–200 μg/mL) or Leu (50 μM) was changed every 2 days until day 4. A stock solution of PWBPH and Leu was prepared in distilled water. Control cells were treated with distilled water only. On day 4, 3 different views per well were acquired using a light microscope equipped with a CCD camera (BX-51; Olympus, Tokyo, Japan). The diameters of 6 myotubes per view were analyzed using

Image J software (National Institutes of Health, Bethesda, MD, USA). The levels of the myotube diameters were normalized relative to the control group.

### 2.6. Western Blotting

Following treatment, isolated Gas muscle and C2C12 cells were lysed, and protein concentrations were measured using a protein assay kit (iNtRON, Biotechnology, Inc., Seongnam, Gyeonggi, South Korea). The lysates were boiled and separated by 10% SDS-PAGE. Proteins were transferred onto PVDF membranes, which were then blocked with 5% skim milk for 1 h and incubated with primary antibodies (1:1000 dilution) for 3 h. The primary antibodies were detected with an HRP-conjugated secondary antibody (1:1000 dilution), and the protein bands were detected using an enhanced chemiluminescence detection kit (BIOFACT Inc., Daejeon, South Korea). The integrated optical density of each protein band was calculated using Image J software. Values were normalized relative to the housekeeping gene $\alpha$-tubulin or to the total protein.

### 2.7. Histological Analysis of Muscle

Gas muscles were collected and immediately fixed in 10% formalin. The fixed tissue samples were embedded in paraffin and cut into sections 4 μm thick for histological analysis. Sections were stained with hematoxylin and eosin and examined under a light microscope equipped with a CCD camera (BX-51; Olympus). Three randomly selected non-overlapping fields of view per coverslip were imaged using identical settings. The outlines of 30 muscle fibers were digitized from each field, and the myofiber cross-sectional area (CSA) was analyzed using Image J software. The levels of the myofiber CSA were normalized relative to the control group.

### 2.8. Statistical Analysis

The data are reported as the means $\pm$ SD. Statistical significance was determined by analysis of variance (ANOVA) followed by the Tukey–Kramer test, with $p < 0.05$ as the level of significance.

## 3. Results

### 3.1. PWBPH Supplementation Increased Muscle Mass and Exercise Performance

Eight-week-old male ICR mice were acclimatized to the experimental facility for 5 days. The animals were randomly divided into 4 groups, including control, PWBPH (250 mg/kg and 500 mg/kg), and Leu (500 mg/kg) (Figure 1A). Leu is well known to increase the expression and activation of mTOR in various tissues, especially muscle, and lead the process of protein synthesis. To evaluate the effects of PWBPH supplementation on physical characteristics, we examined muscle mass and exercise performance in ICR mice. Following treatment with PWBPH and Leu, body weight did not differ among the groups examined (Figure 1B). The Gas muscle was significantly larger in the PWBPH treatment groups than in the controls (Figure 1C,D). We then investigated the phenotypic effect of PWBPH treatment on exercise performance. As shown in Figure 1D, PWBPH and Leu supplementation increased the time to exhaustion in the treadmill test. According to histological sections of Gas muscles stained with hematoxylin and eosin, PWBPH supplementation significantly increased the myofiber CSA in the PWBPH and Leu supplementation groups compared with the controls (Figure 1F,G).

Notably, a lower dose (250 mg/kg) of PWBPH showed more significant effects on muscle mass, exercise performance, and myofiber CSA, compared with 500 mg/kg PWBPH. Similarly, the muscle differentiation markers MyoD and myogenin also showed higher expression after 250 than 500 mg/kg PWBPH treatment (Supplementary Figure S1). These results suggest that PWBPH supplementation increases muscle mass and exercise performance and that 250 mg/kg is the optimal dose in mice.

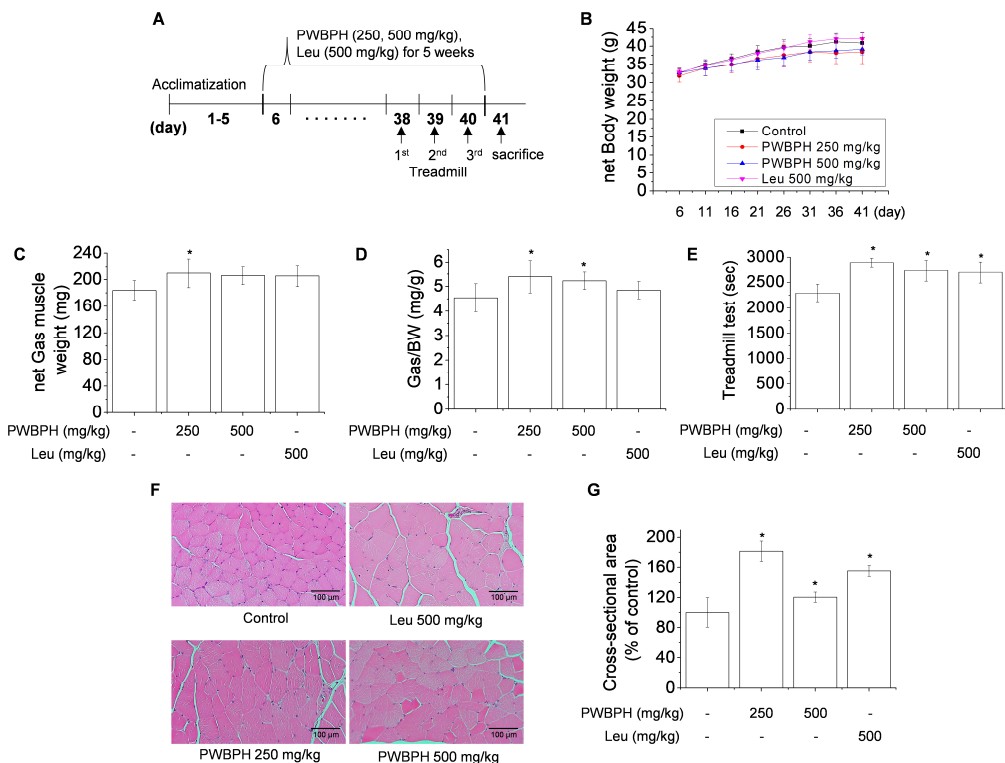

**Figure 1.** Effects of PWBPH supplementation on muscle mass and exercise function. (**A**) Schematic diagram of the experimental protocol in mice. (**B**) Body weights are shown (*n* = 6). (**C,D**) Net gastrocnemius (Gas) muscle weight and Gas muscle/body weight ratios in the 4 groups of mice are shown (*n* = 6). (**E**) Exercise function was determined based on the time to exhaustion in the forced treadmill exercise test (*n* = 6). (**F**) Representative histological images of the Gas muscle stained with hematoxylin and eosin are shown (100× magnification). (**G**) The myofiber cross-sectional area (CSA) was analyzed using Image J software (*n* = 3). Data are shown as the means ± SD. * *p* < 0.05 vs. control group.

### 3.2. PWBPH Supplementation Enhanced Skeletal Muscle Differentiation In Vivo

To obtain a better understanding of the effects of PWBPH on exercise performance in mice, we investigated skeletal muscle differentiation in vivo. As myogenic differentiation is strictly controlled by sequential expression of MRFs, expression of the MRF proteins MyoD and myogenin, and other muscle-specific proteins, such as Pax7 and MYH3, were examined the Gas muscle. PWBPH and Leu supplementation significantly increased the levels of MyoD and myogenin expression (Figure 2). PWBPH also significantly increased the levels of Pax7 and MYH3 expression. However, PWBPH and Leu supplementation did not affect the expression of atrogin-1 and MuRF1, which are E3 ubiquitin ligases involved in protein degradation in skeletal muscle.

### 3.3. PWBPH Supplementation Induced Differentiation and Hypertrophy of C2C12 Cells

To determine whether the differentiation of C2C12 cells was promoted by PWBPH supplementation, the morphological features and protein differentiation were investigated in the late differentiation phase. After exposure to DM alone or DM with PWBPH or Leu for 4 days, the diameter of C2C12 myotube was significantly increased in cells treated with PWBPH and Leu compared with control cells (Figure 3A,B). Expression of MRFs, Pax7, and MYH3 was significantly higher in cells treated with PWBPH and Leu compared with the controls (Figure 3C,D).

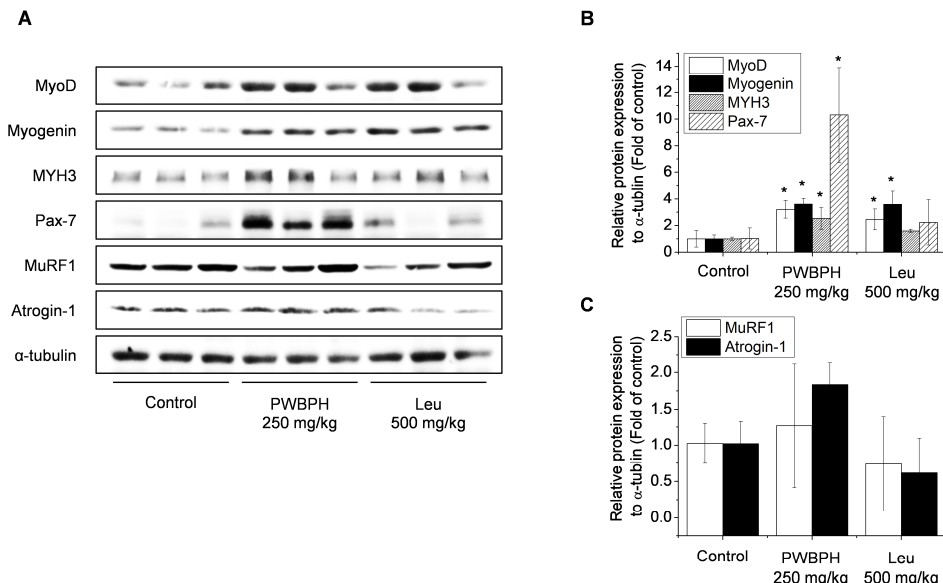

**Figure 2.** Effects of PWBPH supplementation on skeletal muscle differentiation in vivo. (**A**) The levels of muscle differentiation proteins (MyoD, myogenin, MYH3, and Pax7) and muscle degradation proteins (MuRF1 and atrogin-1) in the Gas muscle of the control, PWBPH 250 mg/kg, and Leu 500 mg/kg groups were determined by Western blotting. (**B**,**C**) The band intensities were measured using Image J software. Data are shown as the means $\pm$ SD ($n = 3$). * $p < 0.05$ vs. control group.

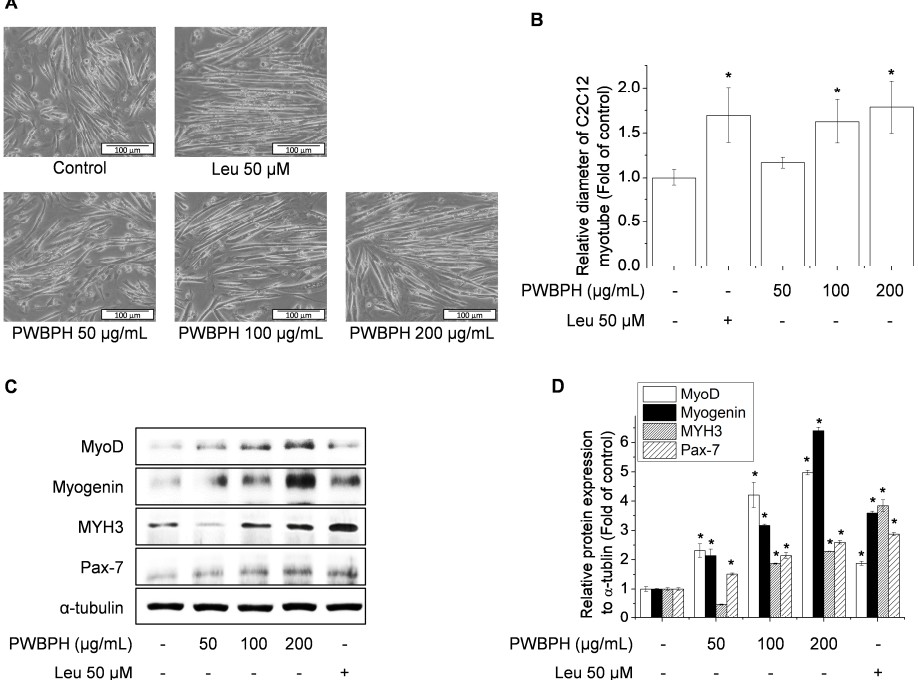

**Figure 3.** Effects of PWBPH treatment on differentiation and hypertrophy of C2C12 cells. C2C12 cells were treated with DM alone, or DM supplemented with PWBPH (50–200 μg/mL), or Leu (50 μM) for 4 days, and were then photographed under a phase-contrast microscope (10× magnification). (**A**) Images are representative of 3 independent experiments (scale bar, 100 μm). (**B**) Myotube diameter was calculated as the average diameter of myotubes using Image J software. (**C**) The levels of muscle differentiation proteins (MyoD, myogenin, MYH3, and Pax7) were determined by Western blotting. (**D**) The band intensities were measured using Image J software. Data are shown as the means $\pm$ SD from three independent experiments. * $p < 0.05$ vs. control group.

*3.4. PWBPH Supplementation Induced Akt/mTOR Signaling Pathways In Vivo and In Vitro*

The Akt/mTOR signaling pathway and its downstream target p70S6K are involved in regulating protein synthesis, and the activities of these enzymes are increased by phosphorylation. PWBPH and Leu supplementation significantly promoted Akt, mTOR, and p70S6K phosphorylation in Gas muscles (Figure 4A,B). In C2C12 cells, PWBPH and Leu treatment significantly increased Akt, mTOR, and p70S6K phosphorylation in a concentration-dependent manner (Figure 4C,D).

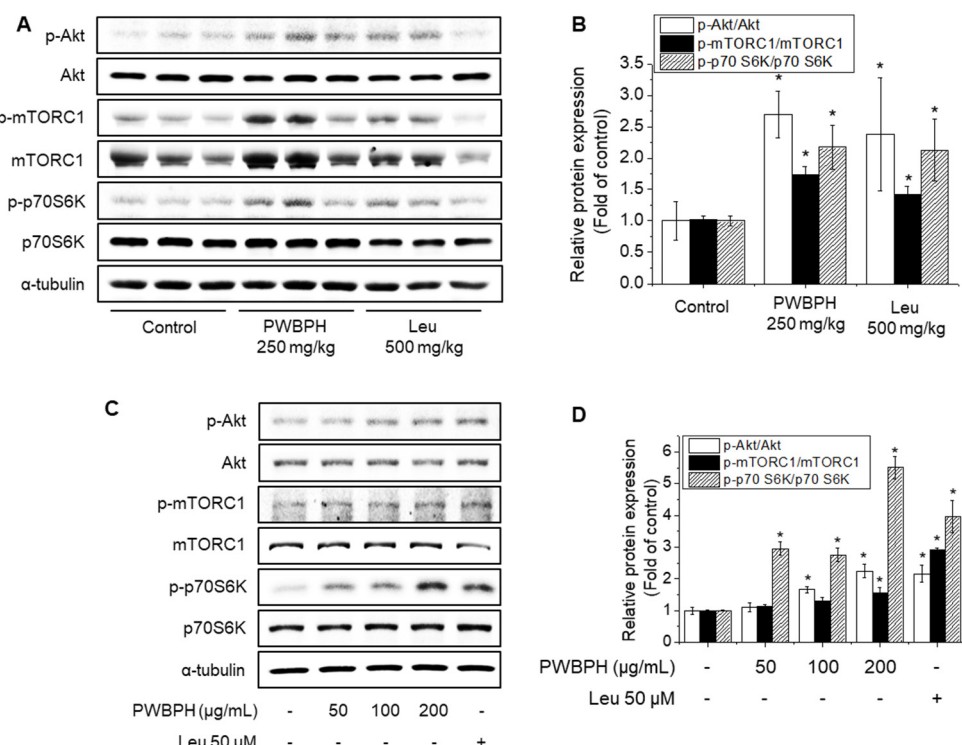

**Figure 4.** Effects of PWBPH treatment on the Akt/mTOR signaling pathway in vivo and in vitro. (**A**) Phosphorylation levels of Akt, mTOR, and p70S6K in the Gas muscle of the control, PWBPH 250 mg/kg, and Leu 500 mg/kg groups were determined by Western blotting. (**B**) The band intensities were measured using Image J software. Data are shown as the means $\pm$ SD ($n = 3$). * $p < 0.05$ vs. control group. C2C12 cells were treated with DM alone, or DM supplemented with PWBPH (50–200 μg/mL), or Leu (50 μM) for 4 days. (**C**) Phosphorylation levels of Akt, mTOR, and p70S6K were determined by Western blotting. (**D**) The band intensities were measured using Image J software. Data are shown as the means $\pm$ SD from three independent experiments. * $p < 0.05$ vs. control group.

## 4. Discussion

BCAA supplementation has beneficial effects on exercise endurance and muscle recovery after injurious exercise [26,27]. Various trials utilizing porcine blood waste by enzymatic hydrolysis have shown significant increases in free amino acids and bioactive peptides, including BCAAs [2,28–30]. However, the effects of PWBPH on exercise function and skeletal muscle protein differentiation remain unknown. In the present study, we demonstrated that chronic administration of PWBPH enhanced exercise function in mice. Interestingly, PWBPH showed greater effects on exercise function and muscle mass in the group administered the lower dose of 250 mg/kg and the Leu administration group (positive control group). Previous studies showed that treatment of BCAAs had weaker effects on biomarkers of muscle gain compared with mixtures of BCAAs and other amino acids in mice and humans [31–33]. These observations suggest that PWBPH enhances exercise function at the optimal dose, and its effect is not only dependent on the amount of BCAAs. Skeletal myogenesis is highly regulated by MRFs, including MyoD and myogenin, which

precisely regulate myogenic cell differentiation in a highly complex process involving cell cycle arrest and multinucleated myotube formation [34]. MyoD-null mice in limb skeletal muscles had a regeneration deficient phenotype in vivo as well as a defective differentiation phenotype in vitro [35,36]. Myofiber deficiencies are observed in myogenin-null mice and removing myogenin before embryonic muscle development, suggesting myogenin has a key role in muscle growth [37]. In the present study, PWBPH treatment increased the expression of MRFs, including MyoD and myogenin, both in vivo and in vitro. PWBPH treatment also increased the myofiber CSA of Gas muscle and hypertrophy of C2C12 myotube. A previous study showed that regular resistance exercise or Leu intake had effects on satellite cell activity and skeletal muscle hypertrophy, with induction of MyoD and myogenin expression in rats [38]. Leu was also reported to have a potential role in improving muscle growth and expression of muscle growth-related genes, including MyoD and myogenin in fish [39].

The Akt/mTOR signaling cascade is involved in muscle protein synthesis induced by nutritional interventions and physical activity, such as BCAAs and resistance exercise [8,24]. Muscle protein synthesis is essential for increasing muscle strength and mass [40]. Therefore, exercise performance can be enhanced by the activation of the Akt/mTOR signaling pathway and increased expression of proteins related to power stroke. Denervation or immobilization of mice and rats induced atrophy via increased expression of both atrogin and MuRF in skeletal muscles. Deficiency in either atrogin or MuRF suppressed denervation-induced muscle fiber diameter and skeletal muscle weight loss in mice. In the present study, PWBPH treatment significantly induced the phosphorylation of Akt, mTOR, and p70S6K both in vivo and in vitro. PWBPH did not affect atrogin or MuRF protein expression in mice but increased muscle mass, indicating that the balance between protein synthesis and degradation regulates the maintenance of muscle mass.

In vertebrates, skeletal muscle regeneration occurs via activation of muscle satellite or stem cells expressing Pax7 and MYH3 to repair damaged myofibers [41,42]. PWBPH treatment increased the protein expression of Pax7 both in vivo and in vitro. However, BCAA intake did not significantly affect Pax7 and MYH3 expression in mice. Lim et al. (2018) also showed that climbing exercise after Leu supplementation for 8 weeks increased the number of Pax7-positive cells per myofiber, but that Leu supplementation alone showed no significant effect in rats. Although there are no specific reports regarding the relationship between BCAA and MYH3, our results indicate that the effect of PWBPH on skeletal muscle regeneration is due to the amino acid or peptide combination of PWBPH and not BCAA alone.

## 5. Conclusions

In conclusion, these data provide evidence that PWBPH supplementation induces Gas muscle mass and exercise performance in mice and hypertrophy of C2C12 cells. Additionally, PWBPH treatment increased the expression of muscle differentiation proteins (MyoD, myogenin, MYH3, and Pax7) and Akt/mTOR signaling cascade both in vivo and in vitro. Overall, our data demonstrated that PWBPH supplementation might help enhance exercise function and muscle fiber differentiation.

**Supplementary Materials:** The following are available online at https://www.mdpi.com/article/10.3390/app12010017/s1, Figure S1: Effects of PWBPH supplementation on muscle differentiation.

**Author Contributions:** Conceptualization, S.W.J. and H.G.J.; validation, S.W.J., G.H.L., J.Y.K. and H.G.J.; formal analysis, S.W.J., G.H.L., E.H.H., J.Y.K., Y.P.H., Y.M.C. and H.G.J.; investigation, G.H.L., J.Y.K. and C.Y.K.; resources, S.W.J. and W.C.; data curation, S.W.J., J.H.C., E.H.H., Y.P.H., Y.M.C., W.C. and H.G.J.; writing—original draft preparation, S.W.J.; writing—review and editing, S.W.J., G.H.L. and H.G.J.; supervision, H.G.J. All authors have read and agreed to the published version of the manuscript.

**Funding:** This work was supported by the Korea Institute of Planning and Evaluation for Technology in Food, Agriculture, and Forestry (IPET) through the Agri-Bioindustry Technology Development

Program, funded by the Ministry of Agriculture, Food, and Rural Affairs (MAFRA) (120053022HD030) and the National Research Foundation of Korea (NRF) grant funded by the Korea government (MSIP) (NRF-2020R1A2C1007764).

**Institutional Review Board Statement:** The study was approved by the Animal Ethics Committee of Chungnam National University (202012A-CNU-191).

**Informed Consent Statement:** Not applicable.

**Data Availability Statement:** The data presented in this study are available on request from the corresponding author.

**Conflicts of Interest:** The authors declare no conflict of interest.

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
