# Peer review of "Effects of Porcine Whole-Blood Protein Hydrolysate on Exercise Function and Skeletal Muscle Differentiation"

_applsci, doi:10.3390/app12010017_

Round 1

Reviewer 1 Report

The study aims to measure the muscular effects of treatment with porcine whole-blood protein hydrolysate (PWBPH). Results seem indicate that PWBPH administration increases muscle weight, exercise performance, myofiber cross-sectional area and activation of the Akt/mTor signalling patway in mice skeletal muscles. In addition, C2C12 cells treated with PWBPH exhibited higher myotube diameter than control cells. While the study is of interest because these results suggest that PWBPH could be a bioresource to increase muscle mass and exercise function, the authors must clarify several points and modify the paper. My comments can be found below.

Title :

In the title, the part of the sentence « skeletal muscle protein synthesis » is not appropriate because authors did not actually measure muscle protein synthesis. (see other comments below). Please modify the title.

Abstract :

Lines 24, 25 and 26 : Authors indicated that they investigated the effect of PWBPH on muscle differentiation and physical activity in mice and C2C12 mouse myoblasts. Please be more precise and indicate that you measured exercise performance. Furthermore, please modify the sentence because you did not measure exercise performance in C2C12 cells.

Lines 28-29 : Please modify the sentence because muscle protein synthesis was not actually measured.

Introduction :

Lines 74-75 : Please modify the sentence because muscle protein synthesis was not actually measured. Furthermore, please modify the sentence because you did not measure exercise function in C2C12 cells.

Lines 75-76 : Please modify this sentence and modulate your affirmation by indicating the models in which exercise performance and muscle regeneration were enhanced.

Materials and Methods :

Lines 92 and 93 : Authors indicated that antibodies against PGC-1a, p-AMPK, AMPK and Sirt1 were used but no data were shown and discussed. Why ?

Line 98 : Please indicate the centrifugal force expressed in g.

Could you provide the composition of PWBPH you used ? proteins, lipids, carbohydrates, ash…

Line 111 : How did you administrated PWBPH and leucine to mice ? Oral gavage ? Did you prepare a stock solution in distilled water or physiological saline ? Did control mice received for 40 days an oral administration of the same vehicule you used to dissolve PWBPH and leucine ? Please precise these information in Mat & Met section.

Line 114 : Authors did not indicate the nutritional state of mice when they were sacrified : fed, fasted or uncontrolled nutritional state. This information is very important because if mice were in a uncontrolled nutritional state, all results concerning measurements of activation of Akt, mTOR and S6 kinase in skeletal muscles can not be used and discussed.

Line 118 : Could you provide information about the apparatus you used to measure exercise performance (name, supplier…)

Results :

Gastrocnemius muscle weights are expressed as a ratio to the total body weight of animal. In Figure 1B, we can observe that PWBPH administration tended to decrease mice body weight after 40 days of treatment in comparison with leucine-treated and control mice. Please indicate in a table the gastrocnemius absolute weights and the hindlimb muscle masses (weights of gastrocnemius + quadiceps + tibialis + soleus + plantaris...) of the 4 mice groups.

Did you control the spontaneous food intake of animals ?  Did PWBPH administration modify spontaneous food intake ? What could be the reason of the tendancy of the decrease of body weight in mice treated with PWBPH in comparison with leucine-treated and control mice ?

Line 181 : Please correct the word « comred ».

Figure 2A : MyHC-3 content was measured by Western-blot. What is « MyHC-3 » ? Did you mean MYH3 the embryonic myosin heavy chain that is expressed in regenerating muscle fiber or MyHC3, a slow-type MyHC ? Please refer to Schiaffino et al. Skeletal Muscle (2015) 5 :22.

Could you provide information about the protein content of an adult fast myosin (MyHC-2A, 2X or 2B) in skeletal muscles of PWBPH treated mice and in C2C12 cells ?

Line 215 : This sentence is not clear : C2C12 cells were treated with PWBPH on day 4 or for 4 days ?

Lines 217-219 : Authors indicated that expression of MyHC3 is higher in cells treated with Leu in comparison with control cells. We do not observe this result in Figure 2B.

Figure 3B : Please check the different cell treatments indicated in the figure : « Leu 50µM + PWBPH 200µg/ml » ?

Reviewer 2 Report

This is a well-designed study that seeks to test the important hypothesis that  porcine whole-blood protein hydrolysate (PWBPH) derived from tissue that is typically discarded can have hypertrophic and muscle-strengthening effects. The authors nicely describe a series of experiments that demonstrate that PWBPH promotes muscle hypertrophy in vivo and in vitro, and induces the expression various myogenic and hypertrophic markers, including the Akt/mTOR cascade. 

I only have a few minor suggestions -

Remove "unequivocal" from line 75

Change "such as control" to "including control" on line 169

Please provide a description of Figure 1A in the Methods section. Were the mice treated daily? If so, please clearly state that

Please state how long the C2C12 cells were treated for, and the treatment concentrations, in the Methods section

Round 2

Reviewer 1 Report

The authors answered most of my questions and improved their manuscript as requested.